# Virulence Factors and Pathogenicity Mechanisms of *Acinetobacter baumannii* in Respiratory Infectious Diseases

**DOI:** 10.3390/antibiotics12121749

**Published:** 2023-12-18

**Authors:** Yake Yao, Qi Chen, Hua Zhou

**Affiliations:** Department of Respiratory and Critical Care Medicine, The First Affiliated Hospital, Zhejiang University School of Medicine, Hangzhou 310003, China; yakeyao0930@126.com (Y.Y.); 22118276@zju.edu.cn (Q.C.)

**Keywords:** *Acinetobacter baumannii*, virulence factors, bacterial pathogenesis, respiratory infectious diseases, lung injury, host–pathogen interaction

## Abstract

*Acinetobacter baumannii* (*A. baumannii*) has become a notorious pathogen causing nosocomial and community-acquired infections, especially ventilator-associated pneumonia. This opportunistic pathogen is found to possess powerful genomic plasticity and numerous virulence factors that facilitate its success in the infectious process. Although the interactions between *A. baumannii* and the pulmonary epitheliums have been extensively studied, a complete and specific description of its overall pathogenic process is lacking. In this review, we summarize the current knowledge of the antibiotic resistance and virulence factors of *A. baumannii*, specifically focusing on the pathogenic mechanisms of this detrimental pathogen in respiratory infectious diseases. An expansion of the knowledge regarding *A. baumannii* pathogenesis will contribute to the development of effective therapies based on immunopathology or intracellular signaling pathways to eliminate this harmful pathogen during infections.

## 1. Introduction

*Acinetobacter baumannii* (*A. baumannii*), a Gram-negative coccobacillus, has globally emerged as a threat in healthcare facilities and primarily infects critically ill patients [1]. *A. baumannii* infections manifest most frequently as ventilator-associated pneumonia, followed by bloodstream, skin, and soft tissue infections. This pathogen also causes post-operational or catheter-associated urinary infections in less common situations [2]. The increasing incidence of multidrug-resistant *A. baumannii* (MDRAB) has rendered many antibiotics invalid and results in high mortality due to infections [3]. Given this concerning situation, the World Health Organization (WHO) has categorized MDRAB as a life-threatening pathogen, and the mechanisms facilitating its survival have attracted great interest from researchers and clinicians [4]. Microbial pathogenesis is associated with the capacity to colonize, cause infections, persist in the host, and show resistance to external stresses [5]. Although the role of bacterial persistence and resistance has generally been accepted as a critical virulence strategy of *A. baumannii*, numerous new virulence factors have been identified over the years [6]. In recent years, multiple studies have focused on *A. baumannii* pathogenesis in pulmonary infections and the interactions between this pathogen and lung epithelial cells. However, the complete and detailed process of *A. baumannii* infections in respiratory diseases remains to be further explored. In this review, we aim to synthesize an updated overview of current research progress regarding antibiotic resistance and virulence factors of *A. baumannii*, specifically describing the pathogenic process and specific molecular mechanisms underlying respiratory tract infections caused by *A. baumannii*.

## 2. Antibiotic Resistance of *A. baumannii*

Once thought to be benign, clinical strains of *A. baumannii* were found to exhibit high susceptibility to most first-line antimicrobial drugs in the early 1970s. However, over the past 50 years, it began to show noticeable resistance to commonly used antibiotics at previously unforeseen rates [7,8]. An important reason was the surge of β-Lactam antibiotic usage, which induced the emergence of drug-resistant strains, leading to the administration of alternative anti-infection strategies like carbapenems. Consequently, carbapenem-resistant *A. baumannii* (CR-AB) emerged and diminished the antibacterial effect observed in clinical patients. Additionally, colistin-resistant *A. baumannii* (Col-R-Ab) isolates were also discovered around the world, raising the alarm for global resistance prevention [9,10].

A global resistance assessment from 2004 to 2014 found that approximately 45% of the isolates of *A. baumannii* were multidrug-resistant (MDR) strains, a rate that was much higher than the rates found in other pathogenic Gram-negative bacteria. Notably, higher MDR rates (~70%) were observed in low-income regions such as Latin America and the Middle East [11]. In 2021, Rizk et al. emphasized the widespread dissemination of MDR strains and carbapenem resistance in developing countries [12].

MDRAB possesses various antimicrobial resistance mechanisms. For example, it synthesizes multiple aminoglycoside-modifying enzymes, most of which are responsible for resistance to fluoroquinolones [9]. The Col-R-Ab strain has been attributed to several mechanisms to date. The modification of lipid A and mutations of lpxA/D/C and pmrA/B genes have been identified as the predominant reasons for colistin resistance by affecting the binding affinity of colistin [13,14,15]. Furthermore, mcr genes (mcr-1, mcr-2, mcr-3, and mcr-4.3) carried by plasmids have recently been detected to have a relation to colistin resistance [14,15]. The emergence of MDRAB and Col-R-Ab calls for a closer look at how to develop and implement effective therapy and resistance surveillance in the future [16].

## 3. Virulence Factors

### 3.1. Outer Membrane Proteins (Omps)

Omps, also known as porins, are thought to be crucial for the pathogenic process of *A. baumannii*, as they are involved in serum resistance, cytotoxicity, apoptosis, biofilm formation, and host–cell interactions [6,17]. To date, Omps that have been identified and studied in detail include OmpA, CarO, Omp33, OprD, and OmpW [4,6].

#### 3.1.1. OmpA

OmpA is a β-barrel-shaped protein with a 2 nm pore diameter and a C-terminal domain located in the periplasm. OmpA is anchored to the cell wall with the help of two conserved amino acids associated with the peptidoglycan layer of the cell wall [18,19]. OmpA is thought to be the most important outer membrane protein, and its amino acid sequence is highly conserved among clinical isolates of *A. baumannii* [4,20]. OmpA plays a versatile role in bacterial virulence and pathogenic processes, including adhesion and invasion, autophagy induction, cellular injury, interaction with the host immune system, biofilm formation, and antibiotics resistance [19,21]. OmpA triggers autophagy in host cells in the respiratory tract by activating multiple signal pathways, including the mitogen-activated protein kinase (MAPK)/c-339 Jun N-terminal kinase (JNK) signaling pathway and the mammalian target of rapamycin (mTOR) pathway, which will be discussed further in subsequent sections [21,22]. Moreover, OmpA also mediates the activation of DRP1 (GTPase dynamin-related protein 1), eventually causing cytotoxicity and mitochondrial fragmentation [23]. Gaddy et al. reported that the OmpA of ATCC 19606 contributed partially to biofilm formation on abiotic surfaces and played a critical role for bacterial adhesion to A549 alveolar cells [19].

#### 3.1.2. CarO

CarO (carbapenem-susceptibility porin) is the second-most abundant protein after OmpA among the Omps of *A. baumannii*, and it causes the acquisition of carbapenem resistance when it is lost or mutated, as CarO can induce the influx of imipenem into the cell wall of bacteria [4,24]. Recently, experiments using *A. baumannii* ATCC 19606 indicated that CarO was implicated in *A. baumannii* attachment to epithelial cells and invasion into the bloodstream and different organs in mammals [25]. Sato et al. proposed that CarO delayed neutrophil infiltration into the lungs, and a decreased level of inflammatory cytokines was detected in respiratory tissues, causing bacterial reproduction and deterioration of pneumonia [26].

#### 3.1.3. Omp33

Another important Omp in *A. baumannii* is Omp33, which acts as a channel for the passage of water. In a previous study, Omp33 was found to matter in the fitness and virulence of *A. baumannii*, such as adherence, invasion, and cytotoxicity of respiratory epithelial cells [18]. In addition, it was reported that this cytotoxic protein triggered apoptosis in immune and connective tissues through the activation of caspases and contributed substantially to pathogenicity [5].

#### 3.1.4. OprD/OccAB1

OprD was identified in carbapenem-resistant *A. baumannii* isolates and was found to be responsible for the transport of antibiotics and molecules like amino acids, sugars, and antibiotics [27]. Recently, OprD was discovered to facilitate the formation of high virulence in the carbapenem-resistant *A. baumannii* clone ST2/KL22, and the knock-out of the *orpD* gene led to virulence diminution. However, the same study found that OprD had no contribution to antibiotic resistance, as a similar antibacterial effect was observed in the isolates with or without the *orpD* gene [28]. In 2016, Zahn et al. determined the crystal structure of OprD and renamed it OccAB1. They found that OccAB1 possessed a large channel size and facilitated the shuttle of various molecules, thus possibly contributing to the assimilation of nutrients and *A. baumannii* survival in vivo [27].

#### 3.1.5. OmpW

OmpW is involved in adaptive resistance to various environmental stresses and has cytotoxic activity against host cells [29]. Gil-Marqués et al. found that OmpW deletion in *A. baumannii* impaired the adherence and invasion ability against A549 cells and reduced biofilm formation. In addition, the loss of OmpW did not affect susceptibility to several antibiotics according to another study using a murine peritoneal sepsis model [30]. OmpW was also thought to be an iron regulon and implicated in the iron uptake process and colistin binding of *A. baumannii*. A study proposed that OmpW formed channels in artificial lipid bilayers and interacted with iron and colistin, but it did not lead to colistin resistance [29].

### 3.2. Lipopolysaccharides (LPS)

Lipopolysaccharides (LPS) have been reported to induce the production of chemokines and inflammatory cytokines such as IL-6, TNF-α, IL-1β, and IL-8 [31]. *A. baumannii* does not contain an O antigen; thus, its LPSs are actually lipooligosaccharides (LOS), which are nevertheless still called LPS in many studies. Lipid A, a component of LPS and the major stimulus for Toll-like receptor (TLR) 4, could promote the release of pro-inflammatory cytokines through the TLR-NF-κB signal axis [32]. In addition, LPS was reported to contribute to the serum resistance of *A. baumannii* and provide a competitive advantage in the host [22,32].

### 3.3. Capsular Polysaccharides (CPS)

Capsular polysaccharides are synthesized by the K-locus gene cluster, and the process of envelope biogenesis is regulated mainly by the OmpR–EnvZ system and BfmRS in the two-component system (TCS) [33,34,35]. CPS promotes immune evasion by means of a reduced attachment between the immunogenic components of *A. baumannii* and immune cells in the respiratory tract [35]. Capsule is necessary for the resistance to phagocytosis, antimicrobics, desiccation, disinfectants, and lysozymes of *A. baumannii* [36]. Multiple studies have indicated that increased production of capsular polysaccharides enhanced the virulence of *A. baumannii* in a mouse model [37,38]. As the main component of bacterial mucus, CPS had been proven to contribute to the formation of a mucoid phenotype, which was a recognized hypervirulence factor in *A. baumannii*, forming a barrier protecting the bacteria from outside stresses and reducing the penetration of antibiotics into the cells. The virulence of mucoid *A. baumannii* depended largely on the CPS production regulated by BfmRS and OmpR–EnvZ [39].

### 3.4. Phospholipase

Phospholipase is a lipolytic enzyme that participates in the metabolism of phospholipids, and it could affect the stability of the epithelial cell membrane. *A. baumannii* has two phospholipase C and three phospholipase D enzymes, which possess substrate specificity toward phosphatidylcholine, a component of the eukaryotic membrane. Recently, it was reported that the inactivation of one of the two phospholipase C genes led to a diminution of the cytotoxic effect on host cells [40]. Additionally, the three phospholipase D genes were considered to be involved in controlling epithelial cell invasion, pathogenesis, and serum resistance [41]. Moreover, phospholipases C and D were found to be implicated in hydrolyzing human erythrocytes, thus contributing to iron acquisition [17].

### 3.5. Pili and Motility

Lacking flagellar-based motility, *A. baumannii* could move via two types of motilities, namely, twitching motility and surface-associated motility. The former is conducted by type IV pili, while the latter is an appendage-independent form of motion [42].

#### 3.5.1. Type IV Pili

Type IV pili help *A. baumannii* extend and retract to keep moving on abiotic surfaces, which is called twitching motility [43]. Type IV pili also play a role in DNA uptake, virulence, and biofilm formation [6,24,44]. In addition to twitching motility, surface-associated movement of *A. baumannii* has been observed in some isolates [6,24,45]. However, this type of motility has little connection with type IV pili [46]. A study demonstrated that impairing these two types of motilities or solely surface-associated motility diminished the pathogenicity of the mutants, while the virulence of twitching-deletion mutants showed no change, indicating that surface-associated motility played a crucial role in bacterial virulence but twitching motility did not [42]. Another study found that the *A1S_2813* and *recA* gene contributed to chemotaxis and surface-associated motility of the *A. baumannii* ATCC 17978 strain [47].

#### 3.5.2. The Chaperone–Usher Pilus System

Csu pilus is a type I chaperone–usher pilus system of *A. baumannii*, which is coded by the *csuA/BABCDE* locus. The expression of the *csu* gene is regulated by BfmRS and GacSA in the two-component system. Csu pilus was previously thought to be important for biofilm formation on abiotic surfaces but not necessary on biotic surfaces like the respiratory epithelium [6]. However, a recent study suggested that the Csu pilus of *A. baumannii* belonged to the mannose-sensitive type I pilus family and was involved in biofilm formation as well as adherence to host cells [48]. In addition, other putative chaperone–usher pilus systems have been discovered in recent years. In a previous study, *A. baumannii* ATCC 17978 was reported to have a type I chaperone–usher pilus assembly system, which was encoded by the photo-regulated pilus ABCD (*prpABCD*) operon and regulated by the photoreceptor BlsA. This light-regulated pilus system was observed to enhance bacterial adhesion and biofilm formation both on plastic and the surface of polarized A549 alveolar epithelial cells, mainly manifesting as a remarkably increased production of filaments and cell chains. This study also proposed that the inactivation of *prpA* resulted in reduced virulence in a Galleria mellonella model. These results suggest that ATCC 17978 cells possess a light-sensing and regulatory system that permits them to interact with biotic and abiotic surfaces to persist in the host. However, this mechanism remains to be validated and elucidated further [49].

### 3.6. Iron Acquisition

Iron is a critical nutrient for *A. baumannii* persistence and ability to cause infection in vivo. *A. baumannii* undertakes various iron acquisition strategies to satisfy its demand, including siderophores, outer membrane vesicles, and heme molecules from lysed erythrocytes [50]. Siderophores are small molecules that are high-affinity iron-specific chelators used by most pathogens as a universal nutrition strategy for iron assimilation and to compete with iron-chelating proteins of the host. To date, *A. baumannii* has been found to possess 10 distinct siderophores, while acinetobactin was found to be the major siderophore that is critical to the survival, growth, and virulence of *A. baumannii* [51,52]. The specific isomerization feature of acinetobactin permits it to bind to iron under the acidic conditions of the infectious process. Consequently, *A. baumannii* competes with commensal bacteria in the upper respiratory tract by employing its iron acquisition advantage with the help of acinetobactin, thus facilitating its colonization and dissemination [6].

Iron uptake has been revealed to affect many other virulence factors, including adhesion, motility, and biofilm formation [50]. There are reliable data to show that iron-rich conditions increase the expression of OmpA in *A. baumannii* [53,54]. Iron concentration has been demonstrated to affect bacterial movement by regulating the type I and type IV pilus systems [50]. An experiment indicated that 18% of motility-related genes, many of which are related to the type I and type IV pilus systems, were markedly downregulated under lower iron concentrations [55].

### 3.7. Secretion Systems

To date, five secretion systems have been discovered in *A. baumannii*, including type 1 secretion system (T1SS), T2SS, T4SS, T5SS, and T6SS, which produce proteins to fulfill multiple cell functions like persistence and resistance, host–cell interactions, and regulation of virulence [17,45].

#### 3.7.1. T1SS

T1SS is composed of an inner membrane ATP-binding protein, a periplasmic adaptor, and an outer membrane pore. It exports proteins, particularly the RTX protein and biofilm-associated protein (Bap), which are related to biofilm formation, maintenance, and bacterial adherence to the respiratory epithelium [50,56,57]. Furthermore, several *A. baumannii* clinical isolates, including the urinary isolate UPAB1 but not ATCC19606, were reported to persist and replicate inside the vacuoles of macrophages, and this process was proven to have a relation to T1SS. The downstream effectors of T1SS in this pathogenic process have been preliminarily screened by researchers but remain to be determined in the future [58].

#### 3.7.2. T2SS

T2SS secretes numerous virulence factors such as lipases (LipA and LipH), alkaline phosphatases, metalloproteinase CpaA, elastases, and phospholipases. Recently, T2SS has been found to participate in the lipid assimilation, serum resistance, and colonization progress of *A. baumannii* [24,59]. The γ-glutamyl transferase enzyme (GGT) was reported to contribute to *A. baumannii* systemic pathology and modulate the immune response of the host. It is secreted by T2SS, inducing apoptosis, interacting with the immune system, and inhibiting the activity of CD4-positive T cells [60]. Another study showed that GGT mediated the inflammatory pathological process and promoted the colonization of *A. baumannii* in host tissues. The same study indicated that the *A. baumannii* isolate AB5075 caused more severe infection, stronger immune response, and heavier tissue damage, which was attributed to its higher extracellular GGT activity. In addition, a higher serum level of GGT was associated with worse lung function in COPD patients [61]. InvL is an adhesin exported by T2SS. It can bind to components of the extracellular matrix and promotes adherence to the urothelium. However, whether InvL serves as a virulence factor in the pneumonia model of *A. baumannii* remains to be elucidated [62].

#### 3.7.3. T4SS

T4SS is required in the conjugative transfer of genetic elements like DNA or plasmids. Therefore, this system is involved in the spread of drug-resistant genes among clinical pathogens, particularly *OXA-23* [45,63,64]. There is a dearth of reports to date on the specific molecular mechanisms of T4SS. While T4SS is crucial in the transfer of antimicrobial resistance, its specific effect on host–pathogen interactions needs to be investigated in the future [59].

#### 3.7.4. T5SS

The type V secretion system (T5SS) has been identified to have five subdivisions to date, namely, type Va, type Vb, type Vc, type Vd, and type Ve. Acinetobacter only encodes type Vb (T5bSS, including the proteins AbFhaB, AbFhaC, CdiA, and CdiB) and type Vc (T5cSS, protein Ata) [59]. AbFhaB participates in the attachment to eukaryotic integrin and fibronectin, while AbFhaC recognizes and translocates AbFhaB to the cell surface [65]. Contact-dependent inhibition (CDI) is a survival strategy of *A. baumannii* to compete with other pathogens. CdiB is a component of the outer membrane pore and mediates the toxic CdiA to pass through the cell membrane to inhibit the growth of neighboring competitors [57]. Additionally, CdiI is an immunity protein in the CDI systems, preventing self-injury of *A. baumannii* from CdiA [66]. The Acinetobacter trimeric autotransporter, or Ata, belongs to T5cSS. Ata is a multifunctional virulence factor in *A. baumannii* that mediates adhesion, invasion, host cell apoptosis, and biofilm formation. Ata mutants were found to present a significant decrease in virulence in in vivo models [17,34,67]. Furthermore, Ata was discovered to mediate a strong immunogenic response and secretion of interleukin (IL)-6 and IL-8, suggesting its crucial role during the infectious process [67,68]. A phylogenetic analysis demonstrated that Ata was identified in 78% of all sequenced *A. baumannii* isolates [67]. Therefore, some studies hold the view that Ata is not an independent virulence factor leading to infections, as the *ata* gene was not found in partial clinical isolates of *A. baumannii* [69].

#### 3.7.5. T6SS

T6SS is activated when stressful stimuli occur and energy costs become higher due to nutrient limitation, cell damage, and bacterial competition [70]. Subsequently, T6SS allows *A. baumannii* to deliver poisonous proteins like nucleases and peptidoglycan hydrolases into neighboring bacteria to outcompete them [71]. Two main proteins, namely, hemolysin co-regulated protein (Hcp) and valine glycine repeat protein (VgrG), have been identified as the core elements of *A. baumannii* T6SS, playing critical roles in the delivery, piercing, and injection of effector proteins [50,72]. Mouse model studies have indicated that T6SS is of importance in the growth velocity, adhesive and invasive competence, and lethality of *A. baumannii*. In addition, *vgrG* gene-deficient *A. baumannii* was not found to be implicated in the formation of biofilm [73]. However, some studies proposed that T6SS is probably not related to virulence since a substantial number of clinical strains of *A. baumannii* lack the assembly proteins for T6SS [74].

### 3.8. GigA, GigB, and GigC

The *gig* (Growth in Galleria) genes were identified by Gebhardt et al. in 2017. GigA and GigB were recognized as virulence determinants in the *A. baumannii* strain AB5075. GigA dephosphorylates GigB and establishes a connection with the nitrogen phosphotransferase system (PTSNtr), a high-conserved metabolic signaling pathway. As a result, GigA and GigB respond to stress signals and promote the transcription of multiple stress response genes. The two regulators are required to cope with external environmental challenges like antibiotic exposure and eventually facilitate infections [75]. Another study compared the similarities and differences of the GigA/GigB stress-sensing pathway between AB5075 and ATCC17978, two commonly studied strains of *A. baumannii*. The researchers reported that *gigA* and *gigB* genes were important for bacterial growth, virulence traits, resistance against macrophages, and G. mellonella infections in both AB5075 and ATCC17978 strains, but they were not necessary for ATCC17978 concerning other stress-resistance phenotypes, including aminoglycoside exposure and Zn2+ toxicity. Additionally, these two genes were proposed to be helpful for antimicrobial resistance in AB5075 [76]. GigC is a LysR-family transcription regulator and contributes to the virulence phenotype of *A. baumannii*. In one study, it was shown that *gigC* mutant strains had a decelerated growth rate in the absence of cysteine because GigC regulated the level of expression of several genes responsible for cysteine biosynthesis and sulfur assimilation [77].

### 3.9. Thioredoxin A (TrxA)

Thioredoxin A (TrxA) has been proven to matter in the regulation of multiple cell functions and virulence in pathogens. In some studies, the colonization level of *A. baumannii* and its lethal rate were clearly decreased in animals exposed to a TrxA-depletion situation, suggesting that TrxA is a potential virulence factor [78]. TrxA is also crucial for resisting oxidative stress and promoting immune evasion by means of modulating cell surface hydrophobicity and the type IV pilus system [79]. Microbial cell surface hydrophobicity (CSH) is linked to virulence, attachment ability to the epithelium, and interactions with the immune system. Thioredoxin A mediates CSH via different approaches, such as disulfide-bond reduction and regulation of the type IV pilus system [80]. Another study using a pulmonary infection model reported that TrxA-deficient strains had better biofilm formation than the wildtype (WT) and were remarkably less toxic than the WT [81].

### 3.10. Polyphosphate Kinase (PPK)

Polyphosphate (PolyP) has been reported to participate in a series of biological functions including bacterial virulence. It is synthesized by polyphosphate kinase (PPK), which comprises the PPK1 and PPK2 families [82]. PPK1 is essential for PolyP synthesis in vivo, and a PPK1-deficient mutant was observed to have attenuated movement since a pilus-like structure was lacking compared to the wildtype *A. baumannii*. The absence of PPK1 also decreased biofilm formation and bacterial persistence in combating outside pressures like antibiotic treatment, oxidative stress, and nutritional deficiency. Moreover, mice that contracted PPK1-depletion *A. baumannii* presented remarkably diminished bacterial concentrations and inflammatory cytokines levels. However, the specific mechanism underlying how PPK induces virulence remains to be further elucidated [83]. The virulence factors mentioned above are summarized in Table 1.

## 4. Pathogenic Mechanisms in Respiratory Infectious Diseases

### 4.1. Attach and Adhere

At the initial stage of infection, *A. baumannii* manages to attach and adhere to extracellular matrix (ECM) proteins and receptors located on the alveolar cellular surfaces [17]. This process depends on the virulence factors described below (Figure 1).

#### 4.1.1. Omps

It is recognized that OmpA has direct interactions with fibronectin on the alveolar and bronchial epithelium surfaces, playing a major role in the adherence to the respiratory epitheliums [18,84]. Additionally, Labrador-Herrera et al. discovered that CarO promoted bacterial adhesion and colonization in mice. Strains lacking CarO manifested impaired adherence and invasion capacity of the human pulmonary epitheliums [13]. ChoP(phosphorylcholine)-containing OprD interacted with platelet-activating factor receptor (PAFR) on the human lung epithelial cell surface; thereafter, it activated a cascade of pathways related to attachment to the alveolar epithelium [85]. Moreover, Omp33 and OmpW were also found to be involved in bacterial adherence, invasion, and cytotoxicity of human lung epithelial cells [5,30].

#### 4.1.2. Protein Secretion Systems

As previously mentioned, AbFhaB(T5bSS) attaches to eukaryotic integrin and fibronectin with the help of its RGD motif [17,65,86]. Ata, which belongs to T5cSS, also has an RGD motif to mediate bacterial adhesion to ECM proteins, and adhesion was found to be remarkably attenuated in Ata-deleted mutants in Galleria mellonella larval models [68]. Another study observed that when the *vgrG* gene, a core element of *A. baumannii* T6SS, was deleted, the adhesive and invasive abilities were weakened [73].

#### 4.1.3. Type IV Pili and CsuA/BABCDE-Mediated Pilus

Type IV pili has been proven to promote adhesion to pharynx and lung carcinoma cells in an in vitro study [87]. Furthermore, previous studies showed that CsuA/BABCDE-mediated pilus was not related to the attachment and adhesion of *A. baumannii* to bronchial epithelial cells [6]. However, recently, Chen et al. first found that Csu pilus belonged to the mannose-sensitive type 1 pilus family and facilitated the adhesion and biofilm formation of *A. baumannii* [48]. MltB, which is classified as a component of the lytic transglycosylase (LT) family, is involved in the remodeling of the peptidoglycan (PG) layer and the release of its fragments. A study proved that inactivation of MltB impaired cell envelope integrity, remarkably decreasing the number of pili on bacteria and their colonization of the respiratory tract [88].

#### 4.1.4. Biofilm-Associated Protein (Bap), Bap-like Protein (Blp), and PstS

*A. baumannii* generates biofilm-associated protein (Bap) that increases cell surface hydrophobicity and, therefore, promotes adhesion to epithelial cells. Lately, two accessory proteins, namely, Bap-like proteins (Blp) BLP1 and BLP2, were found to share a motif at the NH2 terminus with Bap and be co-expressed with Bap. An evident decrease in adhesion to alveolar epithelial cells was detected in *blp1* and *blp2* knock-out mutants, suggesting the complementary functions of Blp with Bap in attachment and adherence [89]. In addition, Gil-Marqués et al. reported that the phosphate sensor PstS was characterized by a highly adhesive and virulent property, but its specific effect on the virulence of *A. baumannii* remained to be elucidated [90].

### 4.2. Internalize and Invade

At the beginning of invasion, *A. baumannii* interacts with platelet-activating factor receptors (PAFRs) via ChoP(phosphorylcholine)-containing OprD, activating a downstream cascade reaction including G protein-coupled phospholipase C(PLC), clathrin, and β-arrestins [91,92]. Adhesion signals from cell receptors are generally related to cellular shapes and motility characteristics. Intracellular signaling pathways induced by the binding of ChoP and PAFRs or the interactions between integrins and particular bacterial adhesins lead to a zipper-like mechanism (receptor-mediated entry) that depends on microfilaments and microtubules, causing local actin cytoskeletal rearrangement at the invasion site and *A. baumannii* internalization. The plasticity of the actin cytoskeleton is thought to be involved in this internalization process. After that, bacteria stay within membrane-bound vacuoles. Proteins like clathrin and β-arrestin-1/2 are then responsible for the direction of vacuoles’ movement as they could link *A. baumannii,* which is coupled with PAFRs, to the vesicular traffic system and deliver it to particular sites. Some studies have proven that this process is essential for the internalization of *A. baumannii* into the respiratory epitheliums [84,85,92,93,94]. Another study showed that *A. baumannii* bound to human carcinoembryonic antigen-related cell adhesion molecule (CEACAM) receptors CEACAM1/5/6, thus enhancing bacterial invasion into the cell membrane and internalization into the vacuoles [95] (Figure 1).

### 4.3. Autophagy, Apoptosis, and Cell Damage

Autophagy is an essential cellular process that degrades senescent organelles, scathed proteins, and intracellular pathogens by clearing them in autophagic lysosomes, which are formed via the fusion of lysosomes and autophagosomes, thus achieving cell metabolism, immune defense, and the renewal of partial organelles [96]. *A. baumannii* could actively induce autophagy but simultaneously block autophagosomes from fusion with lysosomes, making these autophagosomes a niche for its growth and reproduction, thereby escaping its elimination by the immune system and persistently promoting the release of inflammatory cytokines [96].

An et al. proved that OmpA of *A. baumannii* triggered autophagy in macrophages and epithelial cells by activating the mitogen-activated protein kinase (MAPK)/c-Jun N-terminal kinase (JNK) signaling pathway [21]. Another pathway that is implicated in autophagy is the mammalian target of rapamycin (mTOR) pathway. mTOR has been identified to matter greatly in autophagy signal transduction since it integrates various essential biological process signals from upstream, such as energy, nutrition, and redox levels. Recently, OmpA was revealed to promote autophagy by activating the mTOR signaling pathway. OmpA increases the level of activated kinase 1 growth factors (TAK1) and inhibits the activity of mTOR to boost autophagy. Additionally, TAK1 could activate the MAPK pathway, leading to an increase in the release of microtubule-associated protein light chain 3 (LC3) and a decrease in p62, thus promoting autophagy [22]. Furthermore, Ambrosi et al. found that CEACAM5 and CEACAM6 activated the c- JNK1/2-Rubicon- NOX2 pathway, causing LC3-associated phagocytosis (LAP), which led to the formation of autolysosome and eventually microbial degradation [95]. However, given the complexity of the cellular autophagy progress, the details of the host–pathogen interactions of *A. baumannii* remain to be explored in the future. Apart from promoting autophagy, OmpA also mediates the activation of DRP1 (GTPase dynamin-related protein 1), enhancing its accumulation in mitochondria and eventually causing cytotoxicity and mitochondrial fragmentation, increased ROS generation, and cellular damage. Tiku et al. confirmed three *A. baumannii* isolates that had a distant genetic distance, especially the clinical strain AB5075 which caused mitochondrial fragmentation in human lung epithelial cells A549, suggesting its clinical relevance and the conservation of this virulence mechanism [23]. Apoptosis is a process of programmed cell death, and caspases have a crucial role in the regulation of apoptotic signal transduction. Rumbo et al. found that Omp33 induced apoptosis in immune and connective cells through the activation of caspases and modulation of autophagy with subsequent increases in p62 and LC3. In this study, autophagy did not happen in the absence of apoptosis and these processes were proven to generate reactive oxygen species (ROS) [5] (Figure 1).

### 4.4. Incomplete Autophagy

It is recognized that *A. baumannii* prevents or delays the fusion of phagosomes with lysosomes to achieve bacterial survival within host cells. However, the specific molecular mechanism of incomplete autophagy is still elusive. Previous studies found that OmpA played a predominant role in mediating incomplete autophagy during the *A. baumannii* infectious process, as OmpA promoted its colonization in autophagosomes, thus permitting intracellular growth and immune evasion [21,22]. Recently, the newly discovered LncRNA-GAS5/YY1/STX17 molecular network has been elucidated as a regulatory mechanism that controls the disruption of autophagy. STX17, also known as synapsin 17, plays a crucial role in controlling the fusion of autophagosomes and lysosomes. Previous research confirmed that bacteria blocked the formation of autophagic lysosomes through STX17 degradation to evade phagocytic defense. Growth arrest-specific transcript 5 (LncRNA-GAS5) is implicated in the regulation of autophagy and inhibition of STX17 protein expression. Yin Yang 1 (YY1) is a highly conserved zinc-finger transcription factor that enhances the level of STX17. An et al. confirmed that LncRNA-GAS5 and YY1 collectively kept the balance of STX17 concentration by inhibiting each other. As a consequence, autophagy and inflammation remained in a relatively stable state. *A. baumannii* inhibited the expression of YY1, resulting in a surge in the level of LncRNA-GAS5 while STX17 dropped drastically. Moreover, the increased concentration of LncRNA-GAS5, in turn, inhibited the YY1 level, causing a further decrease in STX17 transcription and a reduction in STX17. These interactions led to a disturbance of the balance between inflammation and autophagy [97]. Another study indicated that after *A. baumannii* infected A549 cells, the transcription factor EB (TFEB) was activated and, subsequently, lysosomal biogenesis, autophagy activation, and an increase in A549 cell death were observed. TFEB was proven to matter significantly in the intracellular transport of *A. baumannii* and was necessary for bacterial internalization and survival within host cells. One possible reason is that it contributes to the decrease of lysosomal acidity [85]. Du et al. suggested that YY1 cooperates with TFEB to modulate autophagy by influencing the transcription of genes relating to autophagy and lysosome biosynthesis [98]. These processes cause incomplete autophagy, which allows the persistence and growth of bacteria inside autolysosomes. However, the specific mechanism of exocytosis remains unclear to date (Figure 1).

### 4.5. Inflammatory Response and Cell Death

It is believed that the interference of *A. baumannii* on autophagic clearance is a critical factor that causes the accumulation of inflammatory products and cellular death [21]. To date, several inflammatory response pathways have been proposed and substantial work has been conducted to try to elucidate the detailed mechanisms of the signaling pathways involved in the *A. baumannii* infectious process.

#### 4.5.1. TLR-Nuclear Factor-Kappa B (NF-κB) Signaling Pathway

Toll-like receptors (TLRs) are the leading pathogen recognition receptors (PRRs) implicated in the activation of the innate immune response [99]. Nucleotide-binding oligomerization domain (NOD)-like proteins NOD1/2 are PRRs located in cells sensing peptidoglycan, a specific component of the bacterial cell wall, which trigger a defensive inflammatory response. RIP2 is a molecular adaptor responding to PRRs like NOD1/2 to conduct downstream signaling transduction [34]. The LPS of *A. baumannii* is recognized by PRRs, including TLR2/TLR4, which are located on alveolar macrophages and bronchial epitheliums, as well as NOD1/NOD2 located in the cytoplasm, thus initiating the natural progression of pneumonia by activating receptor interacting protein 2 (RIP2), which has been proven to be crucial in NF-κB activation. The NF-κB signaling pathway regulates the expression of pro-inflammatory cytokines like TNF-α, IL-1β, IL-8, and IL-6, leading to an inflammatory response and facilitating neutrophil recruitment to the lung [34,100,101] (Figure 1). Interestingly, IL-33 treatment has been found to suppress TLR4/NF-κB signaling and decrease the levels of IL-8 and TNF-α, thus inhibiting the systematic inflammatory response caused by *A. baumannii* pneumonia [99]. Another study demonstrated that NOD2 promoted early defense against *A. baumannii* by inducing ROS generation, and bacterial clearance at the early stage helped alleviate pulmonary cell damage from inflammatory cytokine release in the later stage of infections [102]. TLR9 is located intracellularly in endosomes, and it was also found to contribute to defending against *A. baumannii* infections in mouse pulmonary models [50]. Tiku et al. recently found that *A. baumannii* secreted a bioactive lipid that bound to TLR2, causing the activation of the NF-κB pathway in human and murine macrophages. This process produced various pro-inflammatory cytokines like IL-6, IL-8, and TNFα and led to pyroptotic cell death due to the activation of inflammasome signaling, which we will discuss below. In addition, lipase treatment or TLR2 blockage in vitro stopped the signal transduction of inflammation. However, the effect of this bioactive lipid in vivo remains to be examined [103] (Figure 2).

#### 4.5.2. NLRP3 Inflammasome-Caspase Pathway

Inflammasome is a protein complex belonging to the innate immune system. NOD-like receptor 3 (NLRP3) is a predominantly canonical inflammasome that could respond to numerous pathogens like *A. baumannii* [104]. Previous studies have indicated that the NLRP3-ASC-caspase-1/caspase-11 pathway is responsible for producing pro-inflammatory cytokines. Initially, various PRRs like LPS and damage-associated molecular patterns, such as the release of cathepsins, could activate NLRP3 inflammasome in macrophages. After that, the downstream molecular adaptor apoptosis-associated speck-like protein (ASC) was activated, which was followed by the recruitment of caspase-1, thereby facilitating IL-1β production and maturation, resulting in lung injury. The same studies also found that the NLRP3 inflammasome mechanism facilitated pulmonary protection against infections caused by the clinical isolate AB-8879, while it was not indispensable in coping with the type of strain of AB-19606, suggesting the significance of inflammasome in the clinical pathogenesis of *A. baumannii* [105,106].

Activation of NOD-like receptors (NLRs)–caspase-1 is often called the canonical inflammasome pathway. Recently, caspase-11 has been found to promote the release of IL-1β independent of NLRs, which is called a non-canonical inflammasome pathway. Wang et al. proved that caspase-11 deficiency impaired the clearance of *A. baumannii* at infectious sites and, consequently, the deterioration of lung infections and consolidation could be observed. Furthermore, the LPS of Gram-negative bacteria is identified as a dominant factor in caspase-11 activation, as it is thought to be a cytoplasmic ligand for caspase-11. However, the specific molecular mechanism of the inflammatory response activated by caspase-11 remains elusive [104]. Additionally, Omps have been reported to regulate the production of inflammasome. Li Y. et al. confirmed that OmpA inhibited caspase-1 degradation to trigger the release of NLRP3 inflammasome and induce enhanced inflammation, thus aggravating tissue damage [107]. Omp33 was also discovered to activate NLRP3 inflammasome through mitochondria-derived ROS in mouse macrophages [6]. Moreover, Li, D. et al. found that blocking of the p38 MAPK signaling pathway promoted the transformation of macrophage death from pro-inflammatory pyroptosis to non-inflammatory apoptosis, leading to decreased levels of NLRP3 inflammasome and IL-1β in vivo. As a result, pulmonary inflammation and acute lung injury were also alleviated [108] (Figure 2).

#### 4.5.3. CEACAM and PAFR

*A. baumannii* has been reported to adhere to the human carcinoembryonic antigen-related cell adhesion molecule (CEACAM) receptors, a group of immunoglobulin (Ig)-related glycoproteins that are implicated in various biological functions like adhesion, cell signal transduction, and inflammatory response. Interactions of *A. baumannii* with these receptors induce two distinct signal paths. Firstly, *A. baumannii* binds to CEACAM1 receptors of lung cells and triggers IL-8 release through the extracellular signal-regulated kinase (ERK)1/2 and NF-κB signaling pathways to recruit immune cells for bacterial clearance. Another pathway is via CEACAM-5 and CEACAM-6 activated by *A. baumannii*, which could mediate LC3-associated phagocytosis (LAP) [95,109]. The last class of cellular surface receptor that *A. baumannii* interacts with is the platelet-activating factor receptor (PAFR) located on respiratory epithelial cells. Studies have revealed that the cooperation of PAFRs and ChoP-containing OprD results in the activation of ERK-1/2 and mitogen-activated protein (MAP) kinase, which is related to downstream inflammatory and immune responses [91,92]. Lately, ChoP-PAFR-induced invasion into the bronchial epitheliums was found to trigger the activation of Janus kinase (Jak)-signal transducer and activator of the transcription (STAT) signaling pathway, the occurrence of oxidative damage, and eventually the apoptosis of host cells [6]. As a consequence of the processes above, uncontrollable systematic inflammatory response ultimately happened. Excessive inflammation and immune response triggered by *A. baumannii* infections result in an adverse effect on pulmonary epithelial cells, which is recognized as the leading cause of lung injury (Figure 2).

### 4.6. Immunopathologic Response and Histopathology

#### 4.6.1. Interactions with the Immune System

In the early stage of an *A. baumannii* infection, innate immune cells, especially macrophages and neutrophils, have a considerable role in combating this bacterium. Macrophages have been reported to be activated early and modulate the recruitment of neutrophils [2]. LPS interacts with the macrophage TLR4 and activates the NF-κB pathway. As a result, alveolar macrophages produce a variety of pro-inflammatory cytokines, including IL-6, IL-1b, and TNF-α, and neutrophil chemokines like macrophage inflammatory protein-2 (MIP-2) and cytokine-induced neutrophil chemokine 1, which recruit neutrophils from the bloodstream and adjacent tissues to the lungs. Additionally, mast cells and NK cells are also important cell types that recruit neutrophils to infectious sites [110]. Omps also participate in neutrophil recruitment. In a previous study, it was reported that OmpA and periplasmic protein TonB interacted with respiratory epitheliums, producing an antibacterial peptide LL-37, which acted as an indispensable neutrophil chemokine [34]. In addition, Marion et al. found that *A. baumannii* OMVs facilitated the release of chemokines in mouse lungs by binding to TLR2 or TLR4 and initiating downstream signals, thus promoting neutrophil recruitment [111]. In addition, the inflammasome NLRP3 was reported to recruit neutrophils to the lungs in the late stage of an *A. baumannii* infection through IL-1β, which is recognized as a chemokine for neutrophils [105]. Before the neutrophil recruitment process was completed, macrophages produced nitric oxide and reactive oxygen species (ROS) to inhibit *A. baumannii*. Furthermore, macrophages acted as antigen-presenting cells and initiated T cell immunity against *A. baumannii* [2,112].

Neutrophils play a crucial role in the elimination of *A. baumannii* through ROS release and inhibition of neutrophil extracellular traps (NETs) formation [113]. NADPH oxidase is the major ROS producer in phagocytes. Neutrophils have been found to assemble the NADPH oxidase complex after oxygen burst and produce superoxide, which is converted into H_2_O_2_ by superoxide dismutase and eventually kills *A. baumannii* [114]. Another important pathogenesis is that *A. baumannii* suppresses NETs formation. When neutrophils die, their nuclear contents, such as chromatin, microbicidal proteins, and oxidant enzymes, are released and finally form NETs. NETs are thought to be involved in bacterial killing by trapping and inactivating them. In one study, *A. baumannii* was found to inhibit neutrophil adhesion by reducing the expression of the neutrophil surface molecule CD11a to attenuate NETs formation [115]. Some researchers suggested that this pathogenicity allows *A. baumannii* to escape from host immune attacks [115].

There are other mechanisms through which *A. baumannii* interacts with the immune system. It has been shown that OmpA induces apoptosis in DCs by targeting mitochondria and producing ROS [79]. Rosales-Reyes et al. confirmed that OmpA bound to the complement regulator factor H, thus contributing to complement resistance and inhibiting complement activation [116]. Additionally, OMVs were observed to induce J774 macrophage-like cell death and increase pulmonary permeability. Furthermore, OMVs could migrate through host tissues due to their nanoparticle size and destroy the tight junctions of the epitheliums, thereby transporting virulence factors into host cells nearby [79]. Human pleural fluid (PF), an HSA (human serum albumin)-containing fluid, is a host-derived environmental signal. HSA, as a critical component of PF, has been shown to induce strain-specific pathoadaptive responses. Some studies recently found that PF modulated the cytotoxicity and pathogenic behavior of *A. baumannii* and controlled immune responses by changing the metabolic status [117]. Phenylacetate is a bacterial-driven chemotactic agent. HSA in PF induces a phenylalanine (PA) catabolic pathway that does not rely on the phenylacetic acid route, which reduces neutrophil chemotaxis in in vivo assays and promotes an immune evasion state. Notably, PF also enhances the cytotoxicity of *A. baumannii* against macrophages in murine models [28,118] (Figure 3).

#### 4.6.2. Histopathological Changes and Lung Injury

The interactions between *A. baumannii* and the immune system eventually lead to lung injury. Tansho-Nagakawa et al. observed the accumulation of immune cells in the pulmonary infectious foci of mice 14 days after *A. baumannii* treatment, suggesting that inflammation could last for a long time in the lungs and macrophages and dead neutrophils could not be eliminated in time [119]. Lee et al. proposed that the inability of neutrophils to traverse the biofilm built by *A. baumannii* might lead to their prolonged retention in lung tissues, thereby strengthening ROS-mediated damage to respiratory epithelia cells and affecting their function [120]. Other studies proved that these detained neutrophils degraded and leaked their intracellular proteases into the spaces between alveoli, which rendered extensive alveolar capillary injury and persistent inflammation [22,79,120]. Marion et al. found that OMVs interacted with TLR2 and TLR4 located on mouse alveolar macrophages, increasing the generation of the chemokines CCL2 and CXCL1 and the pro-inflammatory cytokines IL-6, IL-1β, and TNF-a, which led to a substantial inflow of neutrophils into the alveolar spaces, thereby causing destruction to the alveolar walls and obstruction of the respiratory tract in mouse lungs, and eventually causing the consolidation of pulmonary tissue [111]. In pulmonary infection models, histological sections of clodronate-treated lungs showed massive reactive oxygen species (ROS) generation, extensive immune cell infiltration, narrow alveolar spaces, and high bacterial loads [120]. In addition, lipid peroxidation brought cytotoxic products like hydrocarbon polymers and lipid peroxides, which promoted oxidative stress and led to lung tissue damage [61]. The high expression of IL-1β has been evidenced to cause acute lung injury [34]. Xiong et al. proposed that IL-1β reduces the expression of VE cadherin, which is essential for intercellular adhesion of endothelium, and destroys the integrity of tight junctions, causing exudation from vascular vessels and inflammatory alveolar damage [121]. In the late phase of an infection, the exacerbated bacterial burden and dissemination of *A. baumannii* to other organs, in turn, promotes the generation of pro-inflammatory cytokines that mediate mucosal damage of the respiratory tract and inflammatory exudation in the alveolar spaces. Eventually, the accumulation of numerous bacteria, necrotized cell fragments, and exudated fibrinous in the alveoli collectively leads to pulmonary consolidation [111] (Figure 3).

### 4.7. Systemic Dissemination

The death of pulmonary epithelial cells and macrophages leads to the disruption of the lung epithelial barrier in the late stage of infection, which permits *A. baumannii* to have access to the blood and disseminate systemically. However, it has been proposed that dissemination could happen in the early stage of infections because live *A. baumannii* cells could be observed in the kidneys 24 h after the injection of the bacteria into the lungs of mice [119]. OmpA was previously thought to be necessary for colonization and spread to adjacent tissues and contribute to bacteremia in the course of infections [23]. IQGTPase-activating protein 1 (IQGAP1) is a characteristic cytoskeleton protein that facilitates intercellular junctions and cellular migration. A study demonstrated that IQGAP1 dynamically regulated the actin cytoskeleton and promoted OmpA-mediated openness of the epithelial barrier of the lungs, leading to an increase in lung permeability, disruption of the respiratory epitheliums, and subsequent bacterial dissemination [3].

Recent studies have found several new virulence factors that boost dissemination. Metalloproteinase CpaA of *A. baumannii* has been reported to inactivate the blood coagulation factor fXII, which promotes intravascular thrombus formation. Thus, CpaA prevents intravascular microthrombus from blocking *A. baumannii* dissemination during bacteremia [122]. CipA is a novel plasminogen-binding protein. Smiline Girija et al. found that the CipA-deficient strains presented a disability in penetrating a single layer of endothelial cells. The authors also highlighted that CipA mediated the degradation of extracellular matrix proteins (ECM). Therefore, it was required for the efficient dissemination of this bacterium [123]. Tuf was recently identified as a moonlighting protein in *A. baumannii* that could degrade fibrinogen and the complement component C3b to evade the attack of the complement pathway, thus enhancing the virulence of *A. baumannii* during the dissemination process [124]. In addition, Zhang et al. found that persistent outside pressure from the respiratory epitheliums promoted the occurrence of *ptk* mutation-mediated mucoid conversion of *A. baumannii*. The mutant strains presented attenuated adherence and enhanced antiphagocytic capacity by prolonging the length of the capsular exopolysaccharide chain. As a result, the acquired mucoid phenotype facilitated the persistence and spread of bacteria in vivo [125]. Moreover, the competence of *A. baumannii* to form biofilms and resist oxidative stress in the respiratory tract is recognized to be conducive to its systemic dissemination [120].

## 5. Conclusions and Perspective

In the last few decades, *A. baumannii* has been seen as a benign pathogen with lower pathogenicity. Now, we should attach great importance to this pathogen, which adapts constantly to external stimuli, thus rendering formidable outcomes. The necessity of further exploration of its pathogenic and antibiotic resistance mechanisms is emphasized, given the influence of *A. baumannii* on public sanitation and clinical practice. More innovative techniques are required in the future to obtain greater insight into the molecular mechanisms underlying its pathogenic processes. Furthermore, it is crucial to develop antivirulence strategies aimed at these immunopathologic processes and intracellular signal transduction to combat *A. baumannii* infections.

## Figures and Tables

**Figure 1 antibiotics-12-01749-f001:**
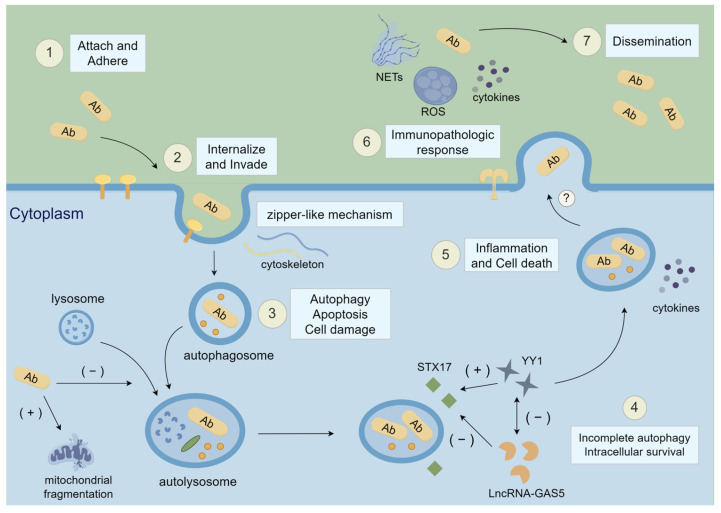
The process of *Acinetobacter baumannii* infection in respiratory diseases. Ab, *Acinetobacter baumannii*; STX17, synapsin 17; YY1, Yin Yang 1; LncRNA-GAS5, growth arrest-specific transcript 5; NETs, neutrophil extracellular traps; ROS, reactive oxygen species. The question mark indicates that the process is hypothetical. The numbers represent the general order of the infection process. This figure was drawn by Figdraw https://www.figdraw.com/ (accessed on 10 November 2023).

**Figure 2 antibiotics-12-01749-f002:**
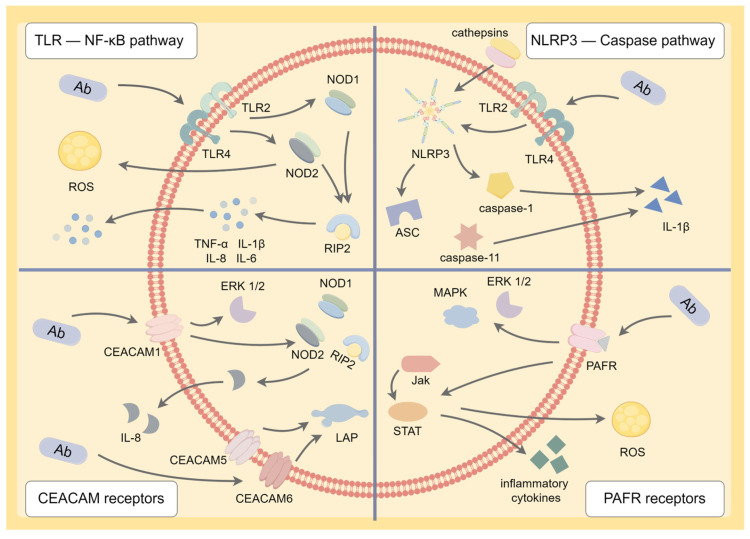
The inflammatory pathways occurred in respiratory epithelial cells and alveolar macrophages during *Acinetobacter baumannii* infection. Ab, *Acinetobacter baumannii*; ROS, reactive oxygen species; NOD, nucleotide-binding oligomerization domain-like proteins; RIP2, receptor interacting protein 2; NLRP3, NOD-like receptor 3; ASC, apoptosis-associated speck-like protein; CEACAM, carcinoembryonic antigen-related cell adhesion molecule; ERK, extracellular signal-regulated kinase; LAP, LC3-associated phagocytosis; PAFR, platelet-activating factor receptor; MAPK, mitogen-activated protein kinase; Jak, janus kinase; STAT, signal transducer and activator of the transcription; This figure was drawn by Figdraw.

**Figure 3 antibiotics-12-01749-f003:**
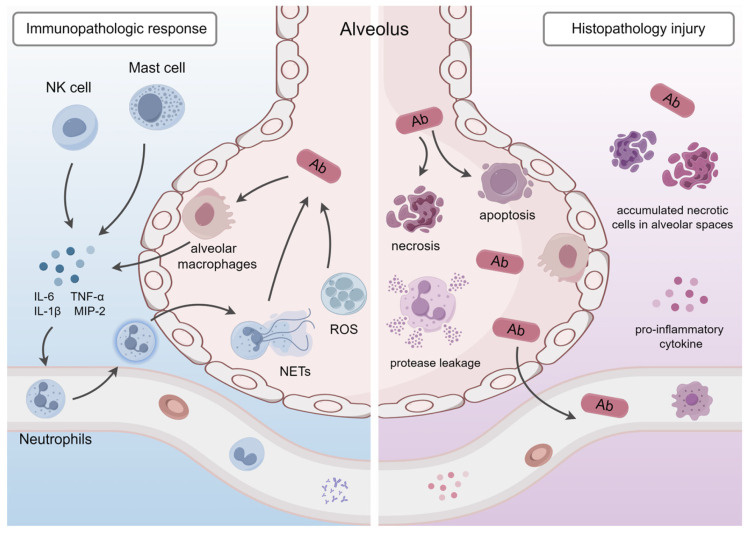
Interactions with the immune system of *Acinetobacter baumannii* and histopathological injury in alveoli. Ab, *Acinetobacter baumannii*; ROS, reactive oxygen species; NETs, neutrophil extracellular traps; This figure was drawn by Figdraw.

**Table 1 antibiotics-12-01749-t001:** Overview of *A. baumannii* virulence factors reported in this review.

Category	Virulence Factor(s)	Function(s)	References
Outer membrane proteins (Omps)	OmpA	Adhesion, invasion, autophagy induction, cellular injury, host–pathogen interactions, biofilm formation, antibiotics resistance	[19,21]
	CarO	Adhesion, invasion, host–pathogen interactions, antibiotic resistance	[4,24,25]
	Omp33	Adhesion, invasion, apoptosis induction	[5]
	OprD/OccAB1	Nutrient intake	[27,28,29]
	OmpW	Iron assimilation, adhesion, invasion, cytotoxicity, biofilm formation	[29,30]
Lipopolysaccharides	LPS	Pro-inflammatory effect, serum resistance	[22,31,32]
Capsular polysaccharides	CPS	Stress resistance, persistence, antibiotic resistance	[33,34,36,37,38,39]
Phospholipase	Phospholipase C/D	Invasion, serum resistance, iron assimilation	[17,40,41]
Pili and motility	Type IV pili	Motility, biofilm formation	[43,44]
	Csu pilus	Biofilm formation	[48]
	Photo-regulated type I chaperone–usher pilus	Biofilm formation, adhesion, virulence	[49]
Iron acquisition	Acinetobactin	Iron assimilation	[50,51,52,53,54,55,56,57,58]
Secretion systems	T1SS	Intracellular survival, adhesion, biofilm formation	[56,57,58]
	T2SS	Adhesion, serum resistance, host–pathogen interactions	[59,60,61,62]
	T4SS	Antibiotic resistance, host–pathogen interactions	[63,64]
	T5SS	Adhesion, invasion, apoptosis induction, biofilm formation	[65,67,68,69]
	T6SS	Stress resistance, killing of bacterial competitors, biofilm formation	[70,71,72,73,74]
Gig (Growth in Galleria)	GigA/B/C	Stress resistance, antibiotic resistance	[75,76,77]
Thioredoxin A	TrxA	Stress resistance, adhesion, host–pathogen interactions, biofilm formation	[79,80,81]
Polyphosphate kinase	PPK	Motility, stress resistance, biofilm formation	[82,83]

## Data Availability

No new data were created or analyzed in this study. Data sharing is not applicable to this article.

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
