# Peer review of "Virulence Factors and Pathogenicity Mechanisms of Acinetobacter baumannii in Respiratory Infectious Diseases"

_antibiotics, 2023, doi:10.3390/antibiotics12121749_

Round 1

Reviewer 1 Report

Comments and Suggestions for Authors

The review article, "Virulence Factors and Pathogenicity Mechanisms of Acinetobacter baumannii in Respiratory Infectious Diseases," demonstrates a good piece of work. However, there are suggested changes for improvement.

One crucial enhancement is the incorporation of the host response to A. baumannii infection within respiratory tissue. This addition will provide a more comprehensive understanding of the interplay between the pathogen and the host.

To enhance the clarity of the presentation, it is recommended that the author summarizes the virulence factors using tables. Additionally, the inclusion of figures is encouraged, as relying solely on text may be less effective in conveying complex information. Visual aids can significantly improve the reader grasp of the content.

Comments on the Quality of English Language

The English is acceptable. few gramatical and syntax mistakes

Reviewer 2 Report

Comments and Suggestions for Authors

The manuscript submitted by Yake Yao et al. provides a comprehensive overview about pathomechanisms of Acinetobacter baumannii.

The manuscrip needs a further round of corrections, since (i) English language use may be improved and the text is written a bit sloppy in respect to formal aspects such as a clear distinction between protein level (capital letter at start) and gene level (in italics) or species names (always in italics).

The manuscript would benefit from some figures (e.g. chapter 2, 3.3, 3.6).

In light of the journal and its readership, a chapter focussing on antibiotic resistant strains may be added.

Comments on the Quality of English Language

A number of unusual sentences, wording, grammar mistakes can be found.

Reviewer 3 Report

Comments and Suggestions for Authors

Some English edits needed, but by and large, it is extremely well written.  I am not making specific English usage or grammar comments as that is best left to editorial staff.

Some sections need more detail and depth.  e.g  OmpA.  The authors list several general roles such as inducing autophagy, cellular damage, and antibiotic resistance.  While these are all true, the readers needs more detail on how this protein accomplishes some of these important aspects of pathogenesis.  I could state the same criticisms for each of the virulence factore lised in section 2.

Lines 98-99 need an English assist AND the TCS regulating transcription of the K-locus needs to be named.

Section 2.2 – the authors list LPS and LOS and then proceed to tell us that there is no LPS, only LOS as there is no O antigen.  This needs to be addressed

Section 2.5 – if A. baumannii lacks flagellar-based motility, this needs to be stated. 

What is the Csu pilus?  Needs definition.

“In addition, the ATCC 17978 strain was discovered to produce light-regulated adhesins. Resultantly, enhanced bacterial adhesion and biofilm formation both on plastic and the surface of polarized A549 alveolar epithelial cells were observed.”

In the passage pasted above, light-regulated adhesins need some context, especially in light of “Resultantly (sic) enhanced bacterial adhesion and biofilm formation…”  Please explain.

Line 201 is not clear. And please explain this system somewhat as the alphabet soup that follows in lines 202-203 are difficult to interpret.  In fact, the entire section on T5SS needs to be made much more clear.

Section 3 is much more well-written and clear and interesting.  The sections headings are more informative and more pleasurable to read.  While this section requires some English usage and grammatical editing, it is informationally much more complete and well-developed.

Comments on the Quality of English Language

Sections 1 & 2 suffer from poor English usage and grammar.  Section 3 is much better

Round 2

Reviewer 1 Report

Comments and Suggestions for Authors

The authors have made substantial revisions to the manuscript, and it is now deemed acceptable for publication.

Comments on the Quality of English Language

English is fine. Minor typos or syntax errors

Author Response

We want to thank you very sincerely for your detailed and helpful review. The English usage and grammatical editing have been improved further.

Reviewer 2 Report

Comments and Suggestions for Authors

The authors responded adequately to my points of criticism.

Author Response

We want to thank you very sincerely for your detailed and helpful review. We hope that this review would contribute to the development of knowledge of A. baumannii pathogenesis.

Reviewer 3 Report

Comments and Suggestions for Authors

This revision has much more detail and is more authoritative.  It is also somewhat easier to read.  The addition of figures and a table is helpful, although in Figure 1, it appears as if A. baumanni bud of in membrane-bound vesicles, but are then shown as free bacteria.  Is there such and exocytosis.  Still, the figures are helpful  Overall a much better paper than the original version!

Comments on the Quality of English Language

some edits required; e.g. secretes not secrets.  But improved

Author Response

It is proven that A. baumanni could replicate in autophagosomes as shown in Figure 1. However, the specific mechanism of exocytosis remains unclear to date. The process of exocytosis shown in figure 1 is hypothetical and we have added some explanations in caption and text to make it clearer and more precise. (Line 478) In addition, the English spelling has been further corrected. Thank you again for your helpful comments.